# Follicle-Stimulating Hormone Alleviates Ovarian Aging by Modulating Mitophagy- and Glycophagy-Based Energy Metabolism in Hens

**DOI:** 10.3390/cells11203270

**Published:** 2022-10-18

**Authors:** Juan Dong, Changquan Guo, Zhaoyu Yang, Yangyang Wu, Caiqiao Zhang

**Affiliations:** College of Animal Sciences, Zhejiang University, No. 866 Yuhangtang Road, Hangzhou 310058, China

**Keywords:** FSH, ovarian aging, mitophagy, glycophagy, chicken

## Abstract

As a predominant hormone in the reproductive axis, follicle-stimulating hormone (FSH) is known as the primary surviving factor for follicular growth. In this study, the alleviating effect of FSH on aging chicken granulosa cells (GCs) was investigated. Results showed that FSH activated mitophagy and relieved mitochondrial edema in D-gal-induced senescent GCs, which was evidenced by an increased number of mitophagosomes as well as increased mitochondria-light chain 3 (LC3) colocalization. Mitophagy activation was accompanied by the activation of the AMP-activated protein kinase (AMPK) signaling pathway. Furthermore, upregulated glycophagy was demonstrated by an increased interaction of starch-binding domain protein 1 (STBD1) with GABA type A receptor-associated protein-like 1 (GABARAPL1) in D-gal-induced senescent GCs. FSH treatment further promoted glycophagy, accompanied by PI3K/AKT activation. PI3K inhibitor LY294002 and AKT inhibitor GSK690693 attenuated the effect of FSH on glycophagy and glycolysis. The inhibition of FSH-mediated autophagy attenuated the protective effect of FSH on naturally aging GC proliferation and glycolysis. The simultaneous blockage of PI3K/AKT and AMPK signaling also abolished the positive effect of FSH on naturally senescent ovarian energy regulation. These data reveal that FSH prevents chicken ovarian aging by modulating glycophagy- and mitophagy-based energy metabolism through the PI3K/AKT and AMPK pathways.

## 1. Introduction

After 480 days of age, high-laying hens undergo a remarkable reduction in egg production. Such a decline in ovarian function is characterized by the decreased quantity and quality of ovarian oocyte reserves. In follicle growth, enormous energy from various substrates is required, including glucose, amino acids, and lipids [1,2]. Within follicles, the oocytes have low glycolytic activity due to a lack of glucose transport receptors, and granulosa cells (GCs) serve as the nursing cells to transfer pyruvate, the glycolytic product, into the mitochondria of oocytes for ATP supplementation [3]. Therefore, the quality and energy status of GCs are potential biomarkers in proper oocyte development. Notably, with the aging of ovaries, the GCs experience a decrease in the mitochondrial respiratory function, and such mitochondrial dysfunction always triggers the accumulation of reactive oxygen species (ROS), further leading to GC apoptosis and impairing ovarian function [4,5]. The aged GCs also present a poor capacity for glucose uptake due to the lower expression of glucose transporter 1 (GLUT1), phosphofructokinase (PFK), and lactate dehydrogenase A (LDHA), and the weakened glycolytic activity reduces the supply of pyruvate, which is the necessary energy substrate for maintaining oocyte maturation and relieving oocyte aging [6,7]. Consequently, relieving the mitochondrial damage and improving the cellular glycolytic activity in GCs have become the most imperative and urgent strategies in alleviating ovarian aging.

Mitochondria autophagy, termed mitophagy, is an organelle-specific form of autophagy that eliminates damaged mitochondria. Mitophagy controls the quality of mitochondria by removing damaged or excessive mitochondria that can generate ROS or cause cell death [8,9]. Recent evidence suggests that mitophagy has been widely investigated in follicular growth and shown to facilitate GC survival due to the clearance of damaged mitochondria or other sources of mitochondrial-derived apoptogenic factors when the GCs are exposed to hypoxia or other energy shortages [10,11]. It is widely known that disorders of ovarian aging are frequently associated with mitochondrial dysfunction in the GCs of follicles, but there is no evidence that activating mitophagy can ameliorate this mitochondrial damage. Similar to general autophagy, light chain 3 (LC3) is also considered a modulator of mitophagy. LC3-I is lipidated to LC-II to induce autophagosome formation, and LC3 lipidation results in an increased LC3-II/LC3-I ratio or LC3-II expression, which is often used as a biomarker for mitophagy in numerous tissues and cells, including oocytes and GCs [12]. Here, the role of mitophagy was investigated in preventing GC aging in chickens.

Besides mitophagy, glycophagy (glycogen-specific autophagy) has also been verified to be involved in cellular energy metabolism through its function of degrading cell glycogen within autophagic vacuoles to trigger the release of free glucose that can be rapidly metabolized by cells for energy requirements [13]. The potential mechanism of glycophagy formation involves the binding of starch binding domain protein 1 (STBD1) and the autophagy-related protein 8 (Atg8) family member, GABA type A receptor-associated protein-like 1 (GABARAPL1). STBD1, which consists of a carbohydrate-binding domain and an Atg8-interacting motif, is localized to the glycogen particles and promotes the trafficking of intracellular glycogen and its degradation through glycophagy by interacting with GABARAPL1, thereby forming a specific mechanism of glycophagy [14,15]. The role of glycophagy has been reported to play an important role in maintaining glucose metabolism under conditions of demand for glucose, such as in neonatal and adult cardiomyocytes, hepatocytes, and other organ cells of newborn animals [13,16]. Glucose is the preferred energy source for ovarian energy metabolism, and previous evidence suggests that a massive accumulation of glycogen particles is observed in ovarian follicles, especially in the GCs of antral and preovulatory follicles before ovulation [17,18]. Glycophagy is a link between cellular glucose homeostasis and autophagy, but it is not clarified whether glycophagy can be involved in the glucose metabolism of follicular growth.

D-galactose (D-gal) is a reducing sugar, which results in excessive ROS production and advanced glycation product (AGE) accumulation in animals in vivo and further induces tissue or cell aging [19,20]. Our previous studies have evidenced that D-gal treatment could induce noticeably aging-related changes in chicken ovarian tissue culture in vitro, such as increasing oxidative stress and reducing the activation of related signaling pathways that are involved in cellular antioxidants [21,22]. Hence, a D-gal-induced aging model in GCs is necessary for experimental studies of ovarian antiaging. Follicle-stimulating hormone (FSH) is a glycoprotein gonadotropin synthesized and secreted by the gonadotropic cells of the anterior pituitary gland. It functions as the primary stimulatory factor for follicular development, GC proliferation, steroid hormone synthesis, and reproductive processes. Our previous research has indicated that exogenous FSH injection in vivo accelerates the development of prehierarchical follicles in low-yield laying chickens involving increased follicular angiogenesis [23]. However, the potential mechanism of FSH involved in the prevention of senescence in chicken ovarian follicles is still unknown. Recently, increasing research has revealed that the protective effect of FSH on GC survival is linked to intracellular autophagy activation, mainly in mitophagy. For instance, in mice, the administration of FSH induces mitophagy signaling in follicular granulosa cells and this process promotes GC proliferation and the expression of genes involved in steroidogenic regulation associated with follicle development [11]. Moreover, other evidence suggests that FSH-mediated mitophagy plays a key role in preserving porcine GC viability against hypoxic damage by removing the damaged mitochondria [10]. In addition, FSH injection in mice in vivo upregulates the expression of hypoxia-inducible factor-1α in ovarian follicles, and the latter-mediated autophagy acts as the glycolysis switch and enhances glucose uptake in mouse GCs [24]. Therefore, these results suggest that autophagy has been an important mediator in the effect of FSH on cellular survival, but whether this link involves the effect of FSH in preventing chicken GC senescence remains elusive.

This study aimed to determine the energy modulation and antiaging effects of FSH on the D-gal-induced premature senescence of GCs and a naturally aging chicken ovarian model, investigate whether mitophagy and glycophagy are responsible for the energy metabolism and anti-aging roles of FSH, and elucidate the mechanisms involving the role of FSH on mitophagy and glycophagy.

## 2. Materials and Methods

### 2.1. Tissues Collection and Morphological Observation

Hyline white hens (*Gallus domesticus*) were raised at a local commercial farm with standard husbandry management conditions. All experimental procedures were approved by the Committee on the Ethics of Animals Experiment of Zhejiang University and conducted in accordance with the Guiding Principles for the Care and Use of Laboratory Animals of Zhejiang University (*ZJU20220085*). Ovaries were collected from D280 (peaking laying) and D580 hens (later laying) and then washed in ice-cold sterile phosphate-buffered saline (PBS) three times to remove the excess blood, respectively. The preovulatory follicles (F6-F1) were counted respectively, and ovarian tissues without follicles over 1 mm in diameter and small yellow follicles (SYFs, 6–8 mm) were immediately snap-frozen in liquid nitrogen for the analysis of biochemical parameters and Western blot or fixed for morphological observation. For H&E staining, ovarian tissues and SYFs of D280 and D580 hens and the cultured ovarian tissues were fixed in paraformaldehyde (4%) over 24 h at 4 °C, then dehydrated, embedded in paraffin, sectioned to a thickness of 5 μM, and then stained with hematoxylin and eosin (H&E) according to a conventional protocol. The stained sections were observed using an Eclipse 80i microscope (Nikon, Tokyo, Japan).

### 2.2. Granulosa Cell Culture and Treatments

SYFs from the ovaries of D280 or D580 hens were collected, washed with ice-cold PBS three times, then peeled using tweezers. Sequentially, the collected granulosa layers were washed in PBS to remove the excess yolk. The granulosa layers were digested using 1 mg/mL collagenase 2 (Gibco, Grand Island, NY, USA) for 5 min at 37 °C, then filtered through 200 mesh cell sieves, and gathered by centrifuging at 1500 rpm for 5 min. After being washed in PBS two times, the GCs were resuspended in the complete DMEM medium supplemented with DMEM high glucose (Hyclone, Tauranga, New Zealand), 5% fetal bovine serum (FBS, Hyclone) and a 0.1% mixture of penicillin–streptomycin (Invitrogen, Carlsbad, CA, USA), then cultured at 38 °C, 5% CO_2_ overnight to ensure cell attachment. For GCs involved in D-gal (HY-N0210, MedChemExpress, Shanghai, China) experiments, the SYF-GCs from D280 hens were treated with the complete DMEM medium (DMEM medium containing 5% FBS and 0.1% mixture of penicillin-streptomycin), which contained different concentrations of D-gal (0, 12.5, 25, 50, 100, 200 mM), and cultured for 12 h or 24 h, respectively. D-gal powder was dissolved in the complete DMEM medium and diluted as described for the experimental design. For GCs in etoposide (HY-13629, MedChemExpress) experiments, cells were cultured with the complete DMEM medium containing etoposide (0, 2.5, 5, 10, 20 μM) for 24 h. For FSH (Ningbo Second Hormone Factory, Ningbo, China) treatment, GCs were first cultured in the complete DMEM medium supplemented with D-gal (200 mM) for 24 h and then treated with the complete DMEM medium containing FSH at the different concentrations (0, 0.001, 0.01, 0.1, 1 IU/mL) for a further 24 h. Based on the evaluation of cell proliferation, 0.01 IU/mL FSH was determined as the optimal concentration for the following formal experiments. For treatment with each specific inhibitor, GCs were pretreated with each of these inhibitors for 1 h after D-gal induction and before the addition of FSH. Dorsomorphin (HY-13418A, MedChemExpress, 100 mM), LY294002 (HY-10108, MedChemExpress, 100 mM), and GSK-690693 (HY-10249, MedChemExpress, 100 mM) were dissolved in dimethylsulfoxide and stored at −80 °C. The final DMSO concertation in the DMEM medium did not exceed 0.1% throughout the experiments. To verify the effect of FSH against ovarian GC aging, the naturally aged GCs (D580) were cultured with FSH (0.01 IU/mL) for 24 h. For the examination of autophagic flux, the LC3 turnover assay was performed. GCs were cultured in a DMEM medium containing FSH for 18 h and then 3-Methyladenine (3-MA, HY-19312, MedChemExpress) for another 6 h.

### 2.3. Organ Culture and Treatments

For organ culture, ovaries of the D580 hens were placed in ice-cold PBS supplemented with a 0.1% mixture of penicillin–streptomycin (Invitrogen). Ovarian cortical blocks (1–2 cm^3^) were separated from the surface of the ovaries, placed onto a 0.45 μm Millipore membrane, and then transferred into the individual wells of 24-well plates at 38 °C, 5% CO_2_ with 1 mL DMEM medium (Hyclone) containing with 2 mM glutamine, 10 μg/mL insulin, 5 μg/mL transferrin, 30 nM selenite (ITS medium, Sigma Aldrich, St. Louis, Mo, USA), and 5% FBS (Hyclone). The ovarian cortical blocks were divided randomly into six groups and were treated with FSH (0.01 IU/mL), FSH + LY294002 (10 μM), FSH + dorsomorphin (10 μM), LY294002 (10 μM), and dorsomorphin (10 μM) for 72 h, respectively. After 48 h of culture, bromodeoxyuridine (BrdU, Sigma Aldrich) was added into the complete medium at 20 μg/mL for a further 24 h.

### 2.4. EdU Assay

The 5-ethynyl-2′-deoxyuridine (EdU) assay using an EdU assay kit (Beyotime, Hangzhou, China) according to the manufacturer’s protocol. For the EdU assay, 10 μM EdU was added to the cells, and the cells were incubated for 2 h at 38 °C. The GCs were then fixed with 4% paraformaldehyde for 15 min at room temperature and exposed to 0.5% Triton X-100 for 15 min. After washing three times with PBS, the cells were stained using the EdU assay kit for 30 min. The nucleic acids in all the cells were stained with DAPI (Beyotime, Hangzhou, China) for 5 min. Images were collected by a fluorescence microscope (Carl Zeiss, Germany). All experiments were performed in triplicate.

### 2.5. Cell Viability Assay

The cell viability of cultured GCs was determined by using a Cell Counting Kit-8 (CCK-8, Servicebio, Wuhan, China). In brief, GCs were seeded in 96-well plates and cultured for attachment overnight. After the desired treatments, 10 μL CCK-8 solution was added to each well, followed by incubation for 2 h at 38 °C. An optical density at 450 nm, which indicates a positive correlation with cell viability, was measured using a microplate reader (Varioskan Flash, Thermo Scientific, USA).

### 2.6. Senescence-Associated β-Galactosidase Staining

β-galactosidase staining kit (G1580, Solarbio, Beijing, China) was used to evaluate the senescence of chicken GCs. In brief, the following steps were performed: GCs were first washed twice with PBS and fixed with the β-galactosidase fixative solution for 15 min at room temperature, then washed three times using PBS for 5 min each time, followed by the addition of 1 mL dyeing liquid (10 μL β-galactosidase staining fluid A, 10 μL fluid B, 930 μL fluid C, and 50 μL X-Gal solution), and incubated at 37 °C overnight. Fine blue uniform particles in the cytoplasm were considered as β-galactosidase positive signals.

### 2.7. Mitochondrial Membrane Potential

The mitochondrial membrane potential (MMP) was detected using the mitochondrial membrane potential assay kit (Beyotime) following the protocol. Approximately 1 × 10^6^ cells were seeded in 6-well culture plates and cultured overnight. After treatments, GCs were rinsed twice with PBS, then incubated with 1 mL of JC-1 staining working solution for 20 min at 38 °C. Subsequently, GCs were washed twice with premade JC-1 staining buffer and observed by a fluorescence microscope (Carl Zeiss). Image software was used to analyze the red and green fluorescence intensities and compare the ratio between the average optical densities of each.

### 2.8. Immunofluorescence Staining

The paraffin-embedded sections of D280 and D580 hens’ ovaries, SYFs, and cultured ovaries were deparaffinized, rehydrated, performed with 10 mM sodium citrate buffer (PH 6.0) for 20 min to complete antigen retrieval, and incubated 1% BSA for 20 min at room temperature as the blocking agent. For cellular immunofluorescence staining, GCs were inoculated into 24-well plates and grown on coverslips overnight. After being treated as mentioned above, the GCs were washed using PBS and fixed with 4% paraformaldehyde for 15 min. Then, GCs were washed with PBS three times, permeabilized with 0.5% Triton X-100 for 10 min, and blocked with 1% BSA for 20 min at room temperature. The sections and cells were incubated with GLUT1 (1:100, HUABIO), LDHA (1:200, HUABIO), HK2 (1:200, A0994, ABclonal), BrdU (1:200, G3G4, DSHB, IA), LC3 (1:200, NB100-2220, Novus, Littleton, CO, USA), and TOMM20 (1:100, ET1609-25, HUABIO, Hangzhou, China) overnight at 4 °C. After being washed three times with PBS, the sections and cell coverslips were incubated with the Alexa Fluor 594-conjugated Goat Anti-Rabbit IgG (1:500, AS039, ABclonal, Wuhan, China) or Alexa Fluor 488-conjugated Goat Anti-Rabbit IgG (1:200, AS053, ABclonal) for 1 h at 37 °C in the dark. The TUNEL assay (Vazyme, Nanjing, China) was performed according to the manufacturer’s instructions to detect cell apoptosis. Nuclei were further stained with DAPI (Beyotime, Hangzhou, China) for 5 min. The tissues and cells were observed using a confocal laser scanning microscope (Olympus, Tokyo, Japan) and analyzed by ImageJ software.

### 2.9. Western Blot (WB) Analysis

Ovarian tissues and GCs were homogenized in ice-RIPA lysis buffer (Beyotime) supplemented with phenylmethylsulphonyl fluoride (PMSF) (Beyotime) and centrifuged to collect the supernatant. Protein concentration was detected using a BCA assay (Beyotime). Equal amounts of protein were subjected to 8–12% sodium dodecyl sulfate-polyacrylamide gel electrophoresis (SDS-PAGE) and electrotransferred into nitrocellulose membranes (Millipore, Darmstadt, Germany). The membranes were blocked in 5% milk with Tris-buffered saline Tween-20 (TBST PH 7.4) for 1 h at room temperature, then probed overnight at 4 °C with the following primary antibodies: GLUT1 (1:500, HUABIO), PFKFB2/PFK2 (1:500, ER1915-04, HUABIO), LDHA (1:1000, HUABIO), SDHA (1:500, ET1703-40, HUABIO), Caspase-3 (1:500, ER1802-42, HUABIO), PCNA (1:500, R1306-5, HUABIO), CDK2 (1:500, R1309-3, HUABIO), CCND1 (1:500, RE6025, HUABIO), TOMM20 (1:1000, ET1609-25, HUABIO), AKT1/2/3 (1:500, ET1609-51, HUABIO), Phospho-AKT1 (Ser7473) (1:500, ET1607-73, HUABIO), GSK3 beta (1:500, ET1607-7, HUABIO), Phospho-GSK3 beta (Ser 9) (1:500, ET1607-60, HUABIO), HK2 (1:1000, ABclonal), AMPKa1/AMPKa2 (1:1000, A17290, ABclonal), Phospho-AMPKa1/AMPKa2-T183/T172 (1:1000, AP1171, ABclonal), GABARAPL1 (1:1000, A7790, ABclonal), LC3 (1:500, Novus), STBD1 (1:500, 11842-1-AP, Proteintech, Rosemont, IL, USA), cleaved Caspase-3 (Asp175) (1:500, 9664, Cell Signaling Technology, Beverly, MA, USA), and β-actin (1:5000, EM21002, HUABIO). After being washed, the membranes were incubated with a secondary antibody for 1 h at room temperature. Finally, images were obtained with a ChemiScope 3400 Mini machine (Clinx, Shanghai, China). For Western blot analyses, unless specifically noted, protein levels were calculated from the ratio of corresponding protein/β-actin.

### 2.10. Co-Immunoprecipitation

The whole proteins from GCs were extracted using ice-cold lysis buffer (Beyotime, P0013) for immunoprecipitation. The lysates were centrifuged, and protein concentrations were measured using the BCA assay (Beyotime). A part of the supernatant was used as input control, while the rest was immunoprecipitated overnight at 4 °C by gently rocking with an anti-STBD1 antibody. Approximately 4 μL of antibody was used (400 μg total protein). Protein A/G magnetic beads (Beyotime, P2108) were added to the immunoprecipitates for 2 h with gentle shaking at room temperature. The beads were then washed three times—five times with cold TBS. Immune complexes were denatured for 5 min at 95 °C in 1× SDS-PAGE loading buffer (Beyotime) before immunoblotting analysis.

### 2.11. RNA Extraction and Quantitative Real-Time PCR (qPCR)

The total RNA of GCs was extracted using the Trizol reagent (Invitrogen Co., Carlsbad, USA). An amount of 1 μg total RNA was reverse transcribed using the HiScript II 1st Strand cDNA Synthesis Kit (Vazyme). The qRT-PCR was performed using HiScript II One Step qRT-PCR SYBR Green Kit (Vazyme). The relative quantity of each gene was calculated by the 2^−ΔΔCt^ method and normalized to the endogenous *GAPDH* reference gene. The sequences of the primers are listed in Table 1.

### 2.12. Measurement of Lactate, Pyruvate, and ATP Level

For lactate production in GCs, the culture mediums from different treatment groups were collected and detected with the lactate assay kit (A019-2, Nanjing Jiancheng Bioengineering Institute, Nanjing, China) following the manufacturer’s protocol. For pyruvate measurement, the cultured ovarian tissues and GCs were collected and homogenized in cold PBS. After centrifugation, the supernatants were detected using the pyruvate assay kit (A081-1, Nanjing Jiancheng Bioengineering Institute). The cultured ovarian tissues or GCs were homogenized and lysed in hot water, heated to 95 °C for 10 min, and centrifuged to obtain the supernatant to measure the ATP level using an ATP assay kit (A095-1, Nanjing Jiancheng Bioengineering Institute, Nanjing, China) according to the manufacturer’s protocol.

### 2.13. Measurements of PK, LDH, and SDH Level

Ovarian tissues from D280 and D580 hens were homogenized in cold PBS and then centrifuged to obtain the tissue homogenate supernatant. The supernatants and serum samples were used for the examination of total protein concentration or measurements of the energy metabolism parameters. Total protein concentration, the enzyme activity of Pyruvate kinase (PK, A076-1), lactate dehydrogenase (LDH, A020-2), and succinate dehydrogenase (SDH, A022-1) were measured following the manufacturer’s instructions with the corresponding kits (Nanjing Jiancheng Bioengineering Institute, Nanjing, China).

### 2.14. Transmission Electron Microscopy (TEM)

The GCs were collected and fixed in 2.5% glutaraldehyde overnight at 4 °C. Then, the fixed GCs were post-fixed in buffered 1% osmium tetroxide for 1.5 h, dehydrated in a graded series of ethyl alcohol or acetone, and eventually wrapped in epoxypropane resin. The ultrathin sections at 70–90 nm were cut using an ultramicrotome (Leica EM UC7, Germany), stained with 8% aqueous uranyl acetate and Reynold’s lead citrate, and then observed under a Tecnai G2 Spirit (FEI Company, Hillsboro, OR, USA).

### 2.15. Statistical Analysis

Data were analyzed by one-way analysis of variance (ANOVA) followed by Tukey’s test or Dunnett’s test using the GraphPad Prism software. Results are represented as the mean ± SEM and regarded as significant when *p* < 0.05.

## 3. Results

### 3.1. Senescence-Related Changes in Ovarian Morphology and Laying Performance

Ovarian tissues and SYFs from D280 and D580 hens were performed by H&E staining. Results showed that the number of growing follicles in D580 was lower than that in the D280 group (Figure 1A). Furthermore, the follicular structure of SYFs in D280 manifested an integral structure with a clear boundary between the granulosa layer (GL) and theca layer (TL) and closely arranged GCs, while the SYFs in D580 showed a follicular structure with a loosely arranged GL and TL (Figure 1B). Ten healthy hens from two groups of laying hens were selected randomly for evaluating laying performance. The results showed that there was a sharp decline in hierarchical follicle number from D280 hens to D580 hens (Figure 1C).

### 3.2. Senescence-Related Changes in Chicken Ovarian Energy Metabolism

As shown in Table 2, the activity of PK and SDH in D580 ovarian tissues and serum was significantly lower than that in D280 hens. Notably, levels of LDH in the serum of D580 hens were markedly higher than those in D280, however, no considerable difference was found in ovarian tissues between D280 and D580. To further explore the aging-related changes in ovarian glycolysis, we analyzed the cellular localization and expression pattern of the glycolytic key enzymes in D280 and D580 ovaries. Immunofluorescence results showed that GLUT1, LDHA, and HK2 were located in the GL of both ovarian follicles and SYFs in D280 and D580 hens (Figure 2A–D). Additionally, the mRNA levels of glycolytic enzymes, *GLUT1*, *HK2, PKM*, and *LDHA*, in the GCs of ovarian SYFs decreased sharply with the increasing age of D280 to D580, while no significant differences were found in levels of *HK1* and *LDHB* between the two stages (Figure 2E). Congruously, the Western blot results also revealed that a significant decrease in GLUT1, HK2, PFK2, and SDHA protein levels of both ovarian tissues and SYFs was evidenced from D280 to D580. Interestingly, the expression of LDHA reduced significantly in SYFs during ovarian aging, while no marked difference was found in ovarian tissues between D280 and D580, which is consistent with the changes in ovarian LDH activity (Figure 2F–I). The above results indicate that decreased ovarian glucose metabolism capacity appears with the aging process of hens, and such a decrease is related to poor glycolysis in GCs.

### 3.3. D-Gal Induces Changes in Senescence Markers but Does Not Promote Apoptosis of GCs

To examine the effect of D-gal on GC survival, SYF-GCs from D280 hens were collected and cultured with different concentrations of D-gal for 12 h or 24 h. As shown in Figure 3A, the cell viability decreased significantly with an increasing D-gal concentration. Particularly, 200 mM of D-gal displayed the highest efficiency in reducing the cell viability than other concentrations. Surprisingly, Western blot analysis revealed that, compared with the control group, the GCs exposed to 200 mM D-gal exhibited a lower level of cleaved-caspase-3 (C-Caspase-3)/Caspase-3, indicating that D-gal-induced decline in cell viability was not due to the apoptosis of GCs (Figure 3B,C). Similar to D-gal, etoposide, a DNA-damaging drug, also impairs cell viability in a high-concentration manner (Figure 3D). TUNEL assay confirmed that D-gal treatment did not induce the apoptosis of GCs, whereas etoposide could cause significant apoptosis in GCs (Figure 3E,F). In addition, to test the ability of D-gal in cell senescence induction, we detected the changes in cell proliferation and senescence biomarkers, such as p53 and Lamin B1, in control and D-gal-induced GCs. Analysis of Western blotting showed that, compared with the control GCs, the protein levels of the proliferation-related protein, PCNA, and Lamin B1 decreased significantly in GCs exposed to D-gal treatment, whereas those of p53 increased, in a dose-dependent manner (Figure 3G,H). The effect of D-gal on the proliferation of GCs was also examined by EdU incorporation. Results revealed that the number of EdU-positive cells decreased gradually with the increases in D-gal concentration (Appendix A). It was further observed by β-galactosidase staining that D-gal significantly increased the ratio of β-galactosidase positive cells in a concentration-dependent manner, and 200 mM of D-gal was the optimal concentration for senescence induction in GCs (Figure 3I). Similarly, the mRNA levels of senescence-related genes, *p53*, *p15*, *p21*, and *p16* increased significantly in D-gal-treated GCs compared with that in the control, while the gene expression of *SIRT1* exhibited an opposite trend (Figure 3J).

### 3.4. FSH Alleviates Cell-Cycle Arrest Caused by D-Gal

To determine whether D-gal-induced GC senescence can be reversed, FSH, an efficient hormone in promoting GC survival, was applied to the senescent GCs. As expected, FSH significantly restored the loss of GC viability upon D-gal exposure in GCs (Figure 4A). Consistent with this, FSH treatment also significantly reversed D-gal-induced reduction in the expression of PCNA, CDK2, and CCND1, which represent the proliferation-related markers (Figure 4B–E). Particularly, 0.01 IU/mL FSH was the optimal concentration for reversing the senescence of GCs and was used for subsequent experiments. Furthermore, FSH recovered the decrease in the number of EdU-labeling cells following D-gal treatment and attenuated the damage of D-gal on the proliferation of GCs (Appendix A). D-gal induced cell-cycle arrest in the SYF-GCs of D280 hens. The results showed that most of the GCs were arrested in the G1 phase after D-gal treatment. However, the addition of FSH effectively relieved the cell-cycle arrest induced by D-gal damage (Figure 4F,G). Additionally, RT-qPCR determination showed that FSH reduced the increased expression of senescence-related genes, *p53* and *p21*, caused by D-gal induction in the SYF-GCs of D280 hens, but reversed the decline in the level of the *SIRT1* gene (Figure 4H). Treatment with FSH attenuated the high expression of β-galactosidase activity caused by D-gal induction in GCs (Figure 4I,J).

### 3.5. FSH Attenuates the D-Gal-Induced Mitochondrial Damage of GCs by Activating Mitophagy via the AMPK Pathway

The results of TEM revealed that the mitochondria of D-gal-induced GCs were swollen dramatically, revealing robust mitochondrial injury, while relieved mitochondrial injury and an increased number of mitophagosomes (red arrows) were observed following the addition of FSH (Figure 5A). To better assess mitochondrial dysfunction, the treated GCs were next stained with JC-1, a more specific fluorescent probe for detecting defective mitochondria. As shown in Figure 5B, compared to the control, D-gal treatment induced a significant decline in cellular functional mitochondria. In contrast, FSH partly restored mitochondrial membrane potential, which indicates that FSH-mediated GC survival might be achieved via alleviating mitochondrial damage during the aging process. Western blotting demonstrated that the expression of mitophagic indicators LC3-II and TOMM20 was upregulated in FSH-treated senescent GCs compared with the control and D-gal-treated groups (Figure 5C,D). Moreover, we investigated AMPK, a vital mitophagic modulator, and found that FSH treatment improved the activation of AMPK and increased the phosphorylation of AMPK in senescent GCs (Figure 5E). Congruously, the fluorescence results of LC3 co-localization with TOMM20 revealed that FSH increased LC3 mitochondrial translocation and led to mitophagosome formation compared to control and senescent cells (Figure 5F,G). To determine whether AMPK is associated with the role of FSH in mitophagy, dorsomorphin (Compound C), an AMPK inhibitor, was used to inhibit AMPK expression. As expected, AMPK inhibition completely reversed the increase in the LC3-II and TOMM20 expression of FSH-treated senescent GCs and inhibited the increased mitophagic activity induced by FSH (Figure 5H,I). Furthermore, the increased content of ATP in FSH-treated senescent GCs was also attenuated by the addition of an AMPK inhibitor (Figure 5J).

### 3.6. FSH Upregulates Glycophagy in D-Gal-Induced Premature Senescent GCs via the PI3K/AKT Pathway

The formation of glycophagy in FSH-treated senescent GCs was investigated. As shown in Figure 6A, glycophagosomes enveloping the glycogen particles were observed in senescent GCs upon FSH stimulation (red arrow). Co-immunoprecipitation was performed to explore the combination of STBD1 and GABARAPL1, which is critical for the delivery of glycogen to lysosomes in the autophagy pathway [15]. The protein levels of STBD1 and GABARAPL1 were increased in the FSH-treated senescent GCs compared with those in the control and D-gal-induced senescent GCs (cell lysates) (Figure 6B). Similar results were detected by pulling down STBD1. The binding and interaction of STBD1 and GABARAPL1 were enhanced in the D-gal + FSH group, indicating glycophagy formation. The expression and interaction of STBD1 and GABARAPL1 were accompanied by the upregulation of the PI3K/AKT axis [25,26]. Thus, the mechanisms of glycophagy induced by FSH in D-gal-induced senescent GCs were examined. The protein levels of phosphorylated AKT and phosphorylated GS3Kβ were markedly elevated compared with the control and D-gal-induced senescent GCs (Figure 6C). Importantly, the specific PI3K inhibitor LY294002 and AKT inhibitor GSK690693 abolished the enhanced effects of FSH on the expression of STBD1 and GABARAPL1 (Figure 6D). In addition, the interaction between STBD1 and GABARAPL1 was also attenuated.

### 3.7. FSH Increases Glycolysis of D-Gal-Induced Premature Senescent GCs via PI3K/AKT Activation

We next determined the effects of FSH on the glucose metabolism of premature senescent GCs. As shown in Figure 7A,B, the results of immunofluorescence revealed that FSH significantly enhanced the expression of GLUT1 and LDHA in senescent GCs. Western blot analysis showed that GCs incubated with D-gal exhibited a significant decrease in the levels of glycolysis-related proteins, GLUT1, HK2, and LDHA, whereas FSH abrogated the reduction in these protein expressions following the D-gal treatment (Figure 7C,D). Subsequentially, the products of glucose metabolism, pyruvate and lactate, were also detected in the cell lysates and supernatant, respectively. It was observed that D-gal-induced premature senescent GCs showed lower levels in the production of pyruvate and the release of lactate, while FSH treatment caused a reversal in such a regression of the glucose metabolism (Figure 7E,F). We have verified that the PI3K/AKT axis was involved in the FSH-mediated glycophagy; however, whether this signaling participated in FSH-mediated glucose metabolism was unknown. To test this possibility, LY294002 (PI3K inhibitor) and GSK690693 (AKT inhibitor) were employed to block the PI3K/AKT signaling pathway. Western blot analysis revealed that the inhibition of the PI3K/AKT axis attenuated the enhanced effect of FSH on HK2 and LDHA expression (Figure 7G,H). Following this, the blockage of the PI3K/AKT axis also weakened the promoted effect of FSH on pyruvate production and lactate release (Figure 7I,J). Additionally, immunofluorescent staining for GLUT1 and LDHA further verified the fact that PI3K/AKT signaling is involved in the effect of FSH on glycolysis in senescent GCs (Figure 7K,L).

### 3.8. Inhibition of Autophagy Impairs the Enhanced Effect of FSH on Glycolysis in Naturally Aged GCs

To evaluate whether treatment with FSH was capable of resisting the recessionary glucose metabolism in naturally aging GCs, the GCs of SYFs from D580 hens were treated with or without FSH for 24 h in culture. Firstly, we investigated the role of autophagy in FSH-mediated cell proliferation. As shown in Figure 8A, FSH treatment exhibited a significant increase in cell viability, but such an increase was attenuated by the 3-Methyladenine (3-MA) addition. Moreover, the inhibition of autophagy also attenuated the expression of PCNA in naturally aged GCs following FSH treatment, which indicates that FSH-induced autophagy in GCs was associated with cell proliferation (Figure 8B,C). In accordance with this, 3-MA also significantly attenuated the increased number of EdU-labeled cells under FSH treatment (Figure 8D). To further demonstrate the relationship between FSH-induced autophagy and glycolysis in naturally aging GCs, we detected the products of the glucose metabolism, pyruvate and lactate, of naturally aged GCs when FSH-induced autophagy was blocked. As expected, FSH treatment increased the accumulation of glycolysis products, such as the increase in pyruvate in cell lysates and higher lactate release in the cell culture medium, whereas 3-MA significantly attenuated the effect of FSH on glycolysis (Figure 8E,F). Additionally, the protein levels of GLUT1, HK2, and LDHA in naturally aged GCs were enhanced following FSH treatment but reduced after co-treatment with an autophagy inhibitor (Figure 8G,H). We next investigated the levels of genes involving steroidogenic regulation related to GC growth detected by qPCR. The results showed that autophagy inhibition reduced the higher expression of *CYP11A1* and *StAR* following FSH treatment, which indicates that the FSH-induced autophagy maintained the function of GCs (Figure 8I,J).

### 3.9. Blockage of AKT and AMPK Pathways Weakens the Effect of FSH on Delaying Natural Ovarian Aging

To further clarify the role of AMPK-regulated mitophagy and PI3K/AKT-regulated glycophagy on the action of FSH for delaying chicken ovarian aging, LY294002, a PI3K/AKT axis inhibitor, and Compound C, an AMPK inhibitor, were added into the culture medium. As shown in Figure 9A,B, FSH treatment activated the PI3K/AKT and AMPK signaling and increased the expression of the phosphorylation of AMPK and AKT in naturally aged ovarian tissues, while the addition of inhibitors reversed this activation. The morphology observation of the cultured ovaries was performed by a hematoxylin and eosin (H&E) staining assay. The results showed that compared to the control ovaries, FSH treatment promoted follicular development and maintained the shape of growing follicles with closely arranged granulosa layers and theca layers. In contrast, the inhibition of PI3K/AKT and AMPK pathways induced damage to the granulosa cells, and the shape of growing follicles was also changed, as the granulosa layers were arranged loosely and irregularly (Figure 9C). In addition, through a BrdU incubation assay, we found that a significant increase in the BrdU labeling rate occurred in FSH-treated ovarian tissues compared to that of the control, whereas the inhibition of AMPK and AKT signaling impaired the promoting effect of FSH on ovarian cell proliferation (Figure 9D,E). A TUNEL assay also revealed that FSH treatment markedly reduced the percentage of TUNEL-positive cells in the cultured aged ovaries, but this decline in apoptotic cells was associated with the activation of AMPK and AKT pathways (Figure 9F,G). Moreover, the inhibition of AMPK and AKT pathways significantly downregulated the higher expression of PCNA by FSH treatment (Figure 9H). To investigate the effect of FSH on ovarian energy metabolism, the content of pyruvate and ATP in cultured ovaries was detected. FSH treatment increases the content of pyruvate and ATP in cultured ovaries, but these changes were reversed by the blockage of the AMPK and AKT pathways (Figure 9H,I).

## 4. Discussion

The ovary is the most susceptible to aging. An obvious feature of ovarian aging is the decline of ovaries from a vigorous state, which manifests itself as ovarian volume shrinking and secretory function decline as well as the depletion of the ovarian pool of nongrowing follicles. Poor energy supplementation for follicular growth has been an essential factor in the decline in ovarian function. In this study, we found that treatment of FSH (0.01 IU/mL) significantly promoted proliferation and energy metabolism in D-gal-induced aging chicken GCs and delayed cell senescence. This protective effect of FSH was accompanied by enhanced mitophagy and glycophagy flux involving AMPK activation. In addition, STBD1/GABARAPL1 binding and the PI3K/AKT axis were responsible for glycophagy under the regulation of FSH in the D-gal-induced senescent GCs (Figure 10). These findings extend previous knowledge and provide new insights into the potential mechanisms involving the antiaging effect of FSH.

In normally developing follicles, glucose serves as the preferred energy fuel of GC metabolism and is essential for energy production, extracellular matrix formation, and the supply of pyruvate to the oocyte for ATP production [27,28]. However, with ovarian aging, the alteration of the mitochondrial function and energy metabolism in GCs contributes to the imbalance of energy homeostasis, causing follicular atresia before progressive development. For instance, recent evidence indicated that GCs collected from the antral follicles of aged cows had lower cell proliferation activity and mitochondria numbers than the GCs of young cows [29]. The decline in mitochondria numbers is one of the hallmarks of cellular aging. In addition, aged GCs also exhibited a lower capacity in glucose uptake, and impaired glucose metabolism further reduced the production of pyruvate, the necessary energy substrate for delaying oocyte aging [30,31]. In this study, there was a sharp decline in hierarchical follicle numbers from peak laying hens (D280) to later laying hens (D580), which suggested a decreased rate of follicle selection into the preovulatory hierarchy and marked reproductive aging in hens. Different from mammals, ovarian follicle development in chickens depends not only on proliferation and differentiation in follicular cells but also on the rapid growth of follicular mass due to massive yolk deposition. Thus, such rapid follicular development requires large amounts of energy substrates to metabolize into ATP supplements for oocyte maturation and ovulation. In our present study, from D280 to D580, the PK and SDH activity decreased both in chicken serum and ovaries with increasing age, which suggests that the chickens’ declined reproductive capacity was associated with poor follicular energy metabolism. In addition, the sharp reduction in mRNA levels of glycolysis-related genes, *GLUT1*, *HK2*, *PKM*, and *LDHA*, also revealed that granulosa cells had a poor capacity for glucose uptake and metabolism with ovarian senescence in hens. These data indicate that chicken ovarian aging is due to the regression in the glucose metabolism of follicular GCs and provide a hypothesis in which improving glycolysis in GCs may be a potential strategy for delaying ovarian aging.

Growing evidence has indicated that the common hallmarks of senescent cells are characterized by lower cell viability; flattened and vacuolized cell morphology; cell-cycle arrest; the accumulation of senescence biomarkers, such as p53, p16, and p21; the decreased expression of Lamin B1 and SIRT1; and the increase in senescence-associated β-galactosidase activity [32,33]. In this study, our findings indicated that D-gal-induced senescent GCs exhibited almost all the features mentioned above, which was consistent with these previous studies, suggesting that our model was successful and workable. In growing follicles, the fate of GCs (proliferation, differentiation, or apoptosis) is regulated by the complex coordination between gonadotropins and ovarian regulatory factors, such as GnRH, FSH, leukemia inhibitory factor (LIF), fibroblast growth factor (bFGF), PGE2, etc. [34,35,36]. Almost all gonadotropins can promote GC proliferation and inhibit follicular atresia, with FSH displaying the highest efficiency. Consistently, our data revealed that the addition of FSH into the culture medium markedly recovered the reduction in cell viability and proliferation, relieved cell-cycle arrest, attenuated the expression of p53 and p21, but elevated the level of SIRT1 and further decreased the senescence-associated β-galactosidase activity in D-gal-induced premature senescent GCs. Therefore, these findings provide evidence suggesting the role of FSH in preventing chicken GCs from premature senescence and the mechanisms that need further study.

Mitochondria are abundant organelles in GCs and play a major role in supplying sufficient ATP for cellular survival, proliferation, and differentiation, but these may be directly affected during ovarian aging. For instance, a clinical study showed that for GCs aspirated from the large antral follicles of aged women, the mitochondria exhibited senescence-related changes in their ultrastructure and were accompanied by a decrease in mitochondrial membrane potential (MMP) [37]. Consistently, other studies also demonstrated that there was a significant increase in the proportion of granulosa cells with ruptured mitochondrial membranes in the resting follicles of elder women compared to those of young ones, which was associated with mitochondrial damage with aging [38]. Thus, the poor quality of aging granulosa cells is mainly related to impaired mitochondrial function, suggesting that preventing ovarian aging can be achieved by alleviating mitochondrial damage in GCs. In our study, D-gal treatment induced mitochondrial dysfunction with the changes in mitochondrial ultrastructure and further caused a decrease in the mitochondrial membrane potential in chicken GCs. However, we found that treatment with FSH could restore this mitochondrial dysfunction by maintaining mitochondrial integrity and reliving the abnormal functions of mitochondria in GCs upon D-gal stimulation. Autophagy is a lysosomal degradation mechanism that involves recycling cytoplasmic proteins or damaged organelles to maintain cellular homeostasis. The process of maintaining mitochondrial function and quality is mainly achieved by mitophagy, an organelle-specific form of autophagy [39]. Recent evidence suggests that FSH injection in vivo promotes the activation of autophagy in mouse ovarian GCs, and FSH-mediated autophagy has a protective role in GC proliferation and follicle development through the selective degradation of damaged mitochondria [11]. In addition, further investigation reveals that FSH-induced mitophagy prevents porcine GCs from apoptotic death under hypoxic conditions [10]. Similarly, autophagy is also observed both in the oocytes and GCs of newly-hatched chicken ovaries and serves a positive effect on chicken primordial follicle assembly by regulating glucose metabolism [12]. Following these studies, our report demonstrated that FSH treatment enhanced mitophagy in chicken GCs following D-gal stimulation. The addition of FSH increased the expression of TOMM20 and the LC3-II/LC3-I ratio and enhanced the co-localization of TOMM20 (mitochondria marker) and LC3 (autophagy marker) in senescent GCs. These are the most widely used tests for mitophagy determination.

AMPK is a classic metabolism-related target, and it can mediate autophagy initiation in response to various cellular stresses. A direct link between AMPK and autophagy activation was established when it was shown that AMPK directly phosphorylates ULK1 in at least four residues: Ser467, Ser555, Thr574, and Ser637 [40]. In our study, the activation of AMPK increased with the addition of FSH in D-gal-induced senescent GCs, and p-AMPK expression was further elevated, indicating the enhanced activation of the AMPK signaling pathway in the senescent chicken GCs after the FSH supplement. The inhibition of AMPK abolished the effects of FSH on mitophagy, which was reflected in the reduction in TOMM20 expression, the LC3-II/LC3-I ratio, and cellular ATP levels, indicating that AMPK may be a central regulator connecting FSH and mitophagy. LC3 and TOMM20 are typically used as an indicator of autophagic or mitophagic flux, respectively, and the LC3 mitochondrial translocation (LC3 co-localization with TOMM20) represents the formation of mitophagosomes [41]. AMPK is essential for LC3 lipidation (LC3-I to LC3-II) to induce autophagosome formation, which is consistent with our findings [40].

Here, our results also found that, in addition to increased mitophagic levels, an increased number of glycophagosomes was observed in chicken GCs, indicating promoted glycophagy by FSH treatment. Glycophagy is a combination of glycogen degradation and the autophagy process, which triggers the liberation of free glucose that can be rapidly utilized by cells with energy requirements [42]. The existence of glycophagosomes elevates the content of intracellular free glucose and enhances glucose metabolism, thereby providing the ATP supply for senescent GCs and maintaining their survival. Then, we verified the role of FSH in glycophagy through the PI3K/AKT axis. The interaction of STBD1 and GABARAPL1 was enhanced, and the PI3K/AKT pathway was activated in D-gal-induced senescent GCs after FSH treatment. Furthermore, the loss-of-function analysis suggested that the binding of STBD1 and GABARAPL1 was weakened by a PI3K or an AKT inhibitor. In addition, it was also observed that LY294002 and GSK690693 inhibited the positive effect of FSH in promoting the phosphorylation of Akt at the same level, as well as GSK3β, suggesting that the PI3K/AKT/GSK3β axis is responsible for the FSH-regulated glycophagy in D-gal-induced senescent GCs. Further studies revealed that the inactivation of PI3K/AKT also impaired the elevated glycolysis in senescent GCs with FSH treatment, indicating that glycophagy is critical for the protective role of FSH in energy homeostasis in senescent GCs. Meanwhile, the inhibition of FSH-mediated autophagy weakened the cell proliferation, glycolysis, and mRNA levels of steroidogenic genes in naturally aging chicken GCs. Moreover, the blockage of the PI3K/AKT and AMPK pathways also attenuated the protective role of FSH on the naturally aging ovaries, which display lower ovarian cell proliferation, ATP levels, and pyruvate production, further indicating that FSH prevents chicken ovarian aging by improving energy metabolism through activating the PI3K/AKT and AMPK pathways, which corroborates the results of FSH in delaying aging GCs.

In conclusion, our current study demonstrates that FSH recovers the cellular injury, mitochondrial damage, and energy metabolism disorders in the D-gal-induced premature senescent GCs of laying hens by regulating mitophagy to maintain mitochondrial homeostasis and by modulating glycophagy and glycolysis to increase glycogen utilization.

## Figures and Tables

**Figure 1 cells-11-03270-f001:**
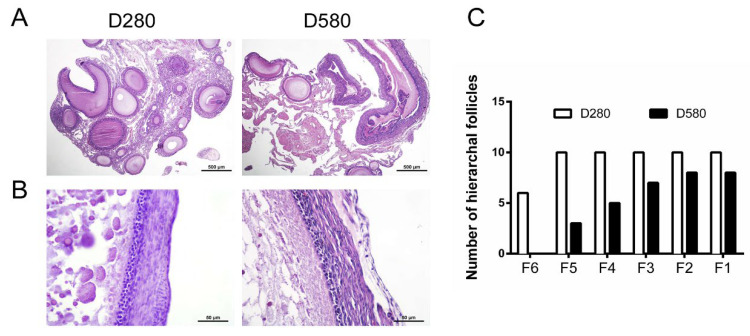
Ovarian histology and the number of hierarchical follicles (F6–F1) in D280 and D580 hens. (**A**) H&E staining of ovarian tissues of D280 and D580 hens. Scale bar: 500 μm. (**B**) Follicular morphology in the SYFs of the D280 and D580 hens. Scale bar: 50 μm. (**C**) The numbers of hierarchical follicles were counted in hens aged 280 and 580 days (n = 10).

**Figure 2 cells-11-03270-f002:**
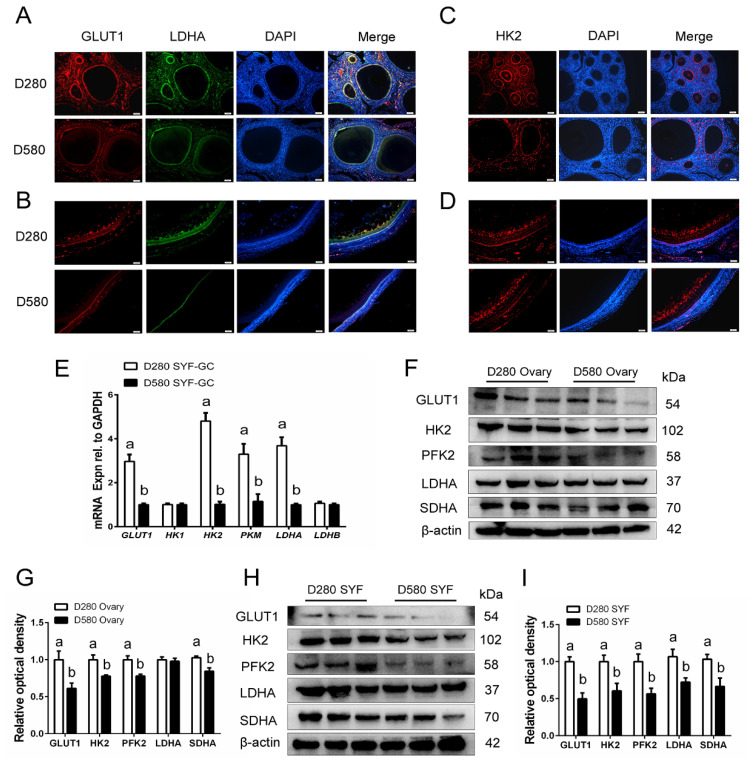
Comparison of glucose metabolism in ovaries and SYFs from D280 and D580 hens. (**A**,**B**) Histological sections of ovarian tissues and SYFs (D280 and D580) were given immunofluorescent labels with GLUT1 (red) and LDHA (green), showing the main distribution in the granulosa layer. Scale bar: 50 μm. (**C**,**D**) Immunofluorescence staining for HK2 in the ovaries and SYFs of hens aged D280 and D580 days. Scale bar: 50 μm. (**E**) Quantitative RT-PCR for *GLUT1*, *HK1*, *HK2*, *PKM*, *LDHA*, and *LDHB* mRNA levels in the GCs of SYFs from hens aged D280 and D580 days. (**F**,**G**) Three hens were randomly selected from each of the D280 and D580 groups; Western blot analysis of GLUT1, HK2, PFK2, LDHA, and SDHA expression in ovaries of hens aged D280 and D580 days. (**H**,**I**) Level of GLUT1, HK2, PFK2, LDHA, and SDHA in the SYFs of hens aged D280 and D580 days. Different lowercase letters indicate significant differences (*p* < 0.05).

**Figure 3 cells-11-03270-f003:**
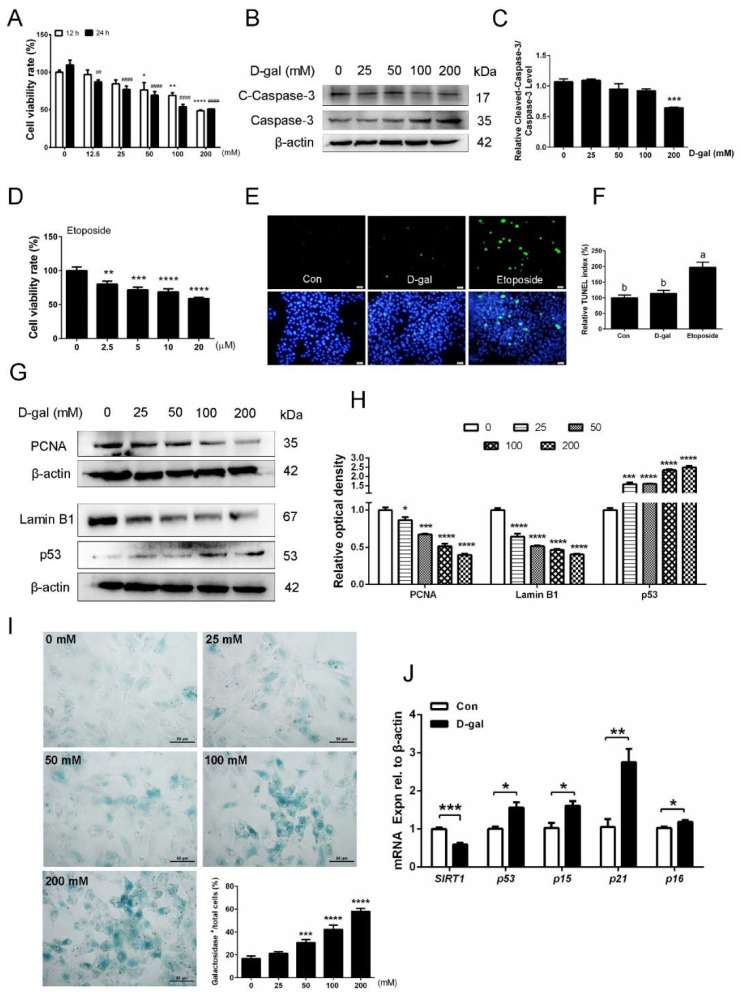
Treatment with D-gal promotes the expression of senescence-associated biomarkers in the SYF-GCs of D280 hens. (**A**) The effect of D-gal on GC cell viability as detected by CCK-8 (n = 6). (**B**,**C**) Western blot analysis of the expression of c-caspase-3 and caspase-3 in GCs exposed to 0 mM, 25 mM, 50 mM, 100 mM, and 200 mM of D-gal for 24 h. (**D**) GCs were treated with etoposide with a concentration of 0 to 20 μm for 24 h and cell viability was measured by CCK-8 (n = 6). (**E**,**F**) TUNEL assay to detect apoptotic GCs in control, D-gal-, and etoposide-treated groups. Scale bar: 20 μm. Different lowercase letters indicate significant differences (*p* < 0.05). (**G**,**H**) Protein levels of PCNA, Lamin B1, and p53 in GCs treated with a gradual increase in the concentration of D-gal. (**I**) Representative images showing β-galactosidase staining in GCs treated with D-gal at 0 mM, 25 mM, 50 mM, 100 mM, and 200 mM concentrations for 24 h. Scale bar: 50 μm. (**J**) qPCR analysis of *SIRT1*, *p53*, *p15*, *p21,* and *p16* mRNA levels in control and D-gal-treated GCs. * *p* < 0.05, ** *p* < 0.01, *** *p* < 0.001, **** *p* < 0.0001 which represent the difference between the groups in which the GCs were treated with D-gal of different concentrations for 12 h compared to the control. # Indicates the difference between 24 h and the control.

**Figure 4 cells-11-03270-f004:**
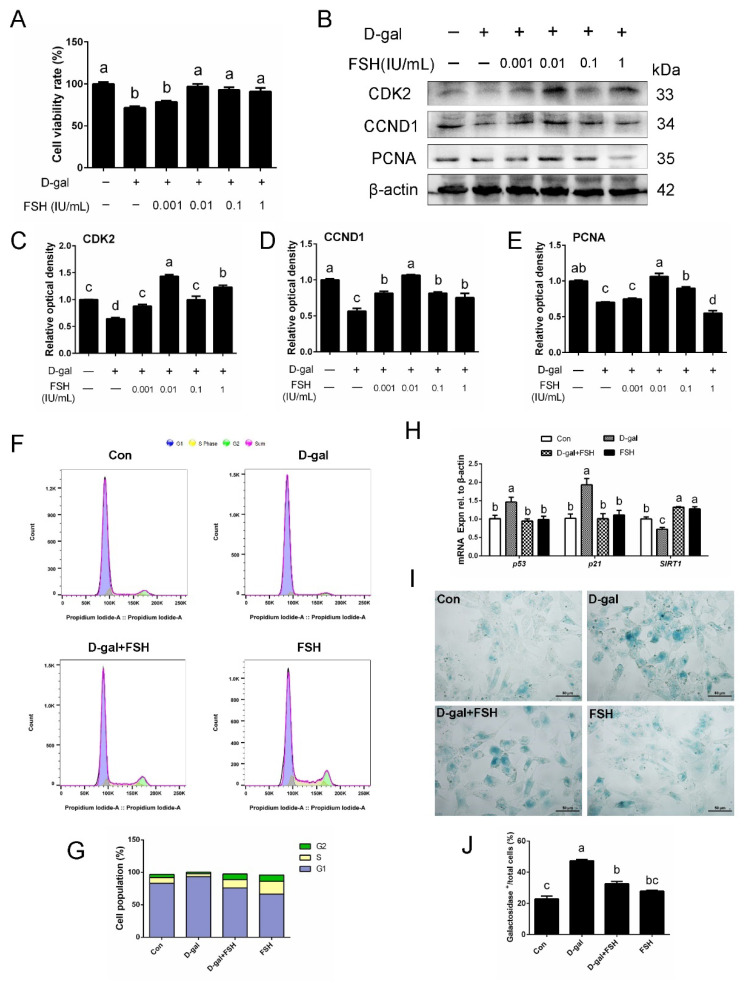
The effect of FSH on the lower cell proliferation and cell-cycle arrest caused by D-gal in SYF-GC of D280 hens. (**A**) SYF-GCs of D280 hens were treated with D-gal (200 mM) for 24 h, and then cultured with FSH at 0.001, 0.01, 0.1, and 1 IU/mL concentrations for another 24 h. Cell viability was measured using the CCK-8 assay. (**B**–**E**) The expression of CDK2, CCND1, and PCNA in cultured GCs was determined by Western blotting. β-actin served as the control for loading. (**F**,**G**) Cell-cycle distribution in control GCs, senescent GCs (treated with 200 mM D-gal for 24 h), senescent GCs (treated with 200 mM D-gal for 24 h) with FSH treatment (0.01 IU/mL for another 24 h), and GCs with FSH treatment alone (0.01 IU/mL) detected using a flow cytometry assay. (**H**) qPCR analysis of *p53*, *p21*, and *SIRT1* mRNA levels in the cultured GCs. (**I**,**J**) β-galactosidase-positive cells/ total cells and representative images of β-galactosidase staining in GCs that were treated as indicated. Scale bar: 50 μm. Different lowercase letters indicate significant differences (*p* < 0.05).

**Figure 5 cells-11-03270-f005:**
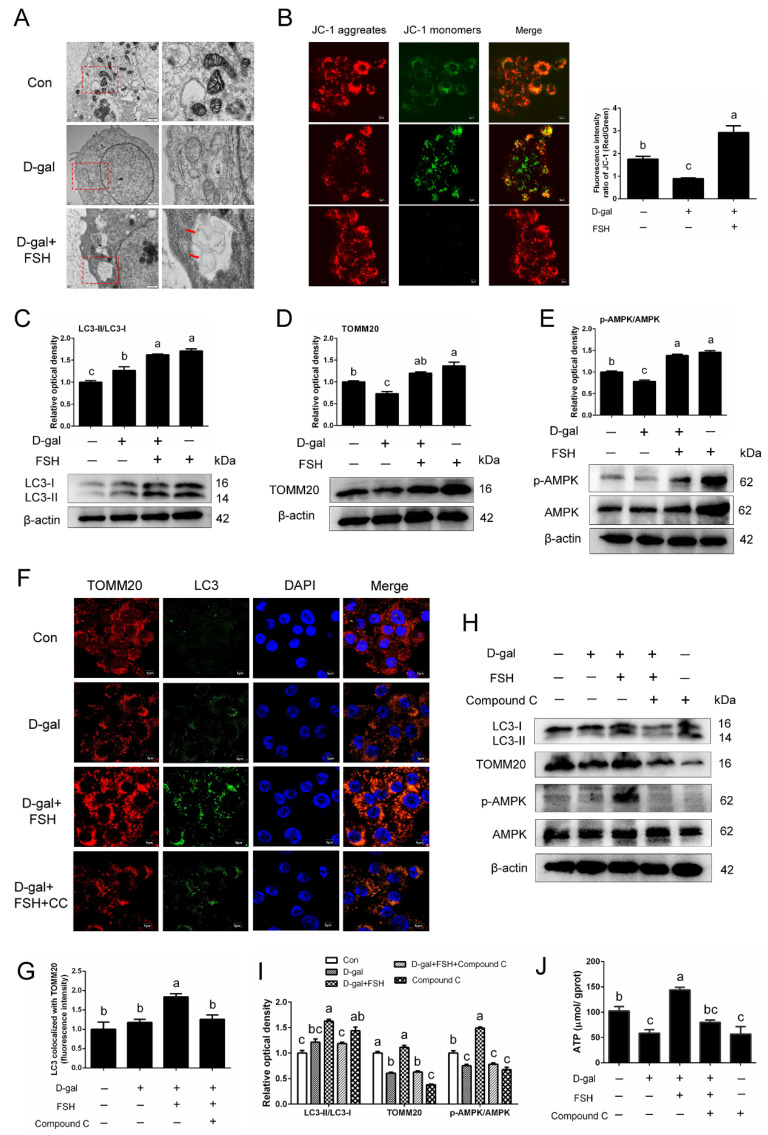
Activation of the AMPK signaling pathway is responsible for FSH-induced mitophagy in senescent GCs. The SYF-GCs of D280 hens were treated with D-gal (200 mM) for 24 h and were induced to become premature senescent GCs. Subsequently, 0.01 IU/mL of FSH was added and stimulated the premature senescent GCs for 24 h. (**A**) Transmission electron micrograph of mitophagosomes (red arrows) in the senescent GCs with FSH treatment. Swollen mitochondria were observed in the D-gal group, whereas this damage was alleviated after FSH addition. Scale bars: 1 μm and 500 nm. (**B**) GCs were collected for JC-1 staining to observe the healthy mitochondria (uptaking red fluorescent aggregates of JC-1) and defective mitochondria (uptaking green fluorescent monomeric JC-1). (**C**,**D**) Western blot and quantitative analyses of LC3-II and TOMM20 expression in normal, D-gal, D-gal + FSH, FSH alone-treated GCs. (**E**) Western blot and quantitative analyses of AMPK phosphorylation in GCs (normal, D-gal, D-gal + FSH, FSH). (**F**,**G**) Colocalization of mitophagic markers (LC3 and TOMM20) in GCs treated as indicated above. CC: Compound C. Scale bar: 5 μm. (**H**,**I**) Western blot analysis of p-AMPK/AMPK, LC3-II, TOMM20 proteins in normal, D-gal, D-gal + FSH, D-gal + FSH +Compound C (AMPK inhibitor), Compound C groups. (**J**) ATP levels of GCs that were treated as indicated above (**H**). Different lowercase letters indicate significant differences (*p* < 0.05).

**Figure 6 cells-11-03270-f006:**
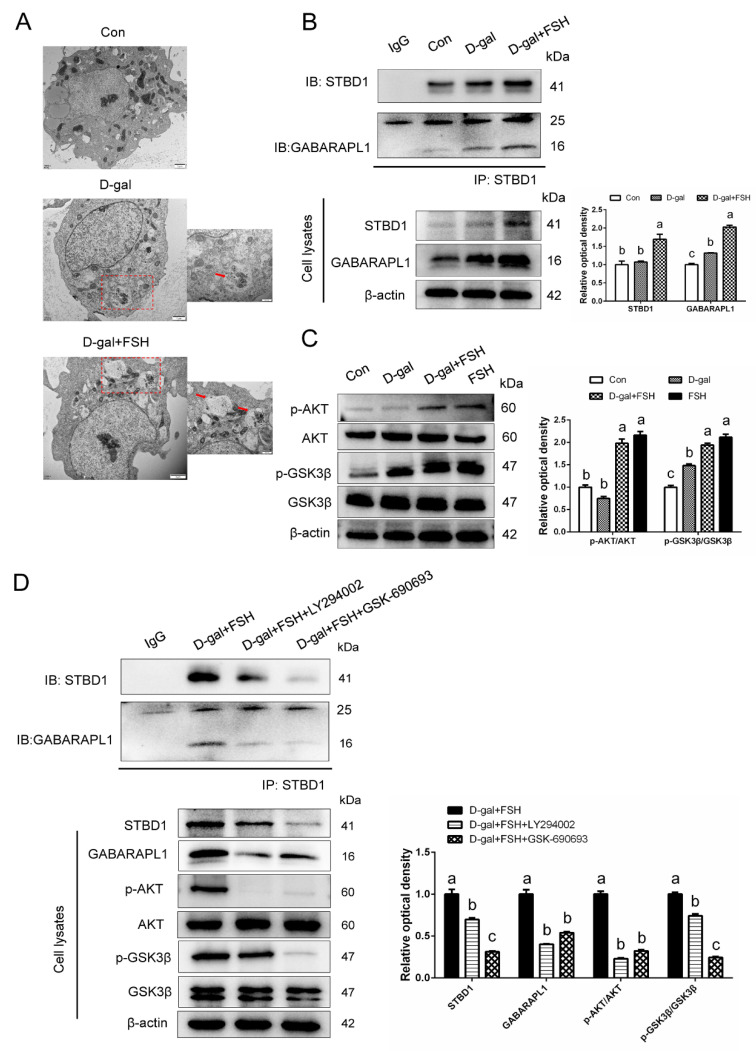
Phosphoinositide 3-kinase (PI3K)/AKT signaling is involved in FSH-induced glycophagy in D-gal-induced senescent GCs. (**A**) Electron micrograph of glycophagosomes (red arrows) in senescent GCs with FSH treatment. The box indicates the area enlarged on the lower right. Scale bars: 1 μm, 500 nm. (**B**) Isolated GCs in the control, D-gal, and D-gal + FSH groups were collected and then immunoprecipitated with STBD1 antibody. STBD1 and GABARAPL1 expression in cell lysates was examined using anti-STBD1 and anti- GABARAPL1 antibodies. (**C**) Western blot and quantitative analysis of AKT and GS3Kβ phosphorylation in the control, D-gal, D-gal + FSH, and FSH alone groups. (**D**) Co-immunoprecipitation of STBD1 and GABARAPL1 in senescent GCs treated with FSH, FSH + PI3K inhibitor LY294002, or FSH + AKT inhibitor GSK690693. The expression levels of AKT and GS3Kβ in cell lysates were also verified. Different lowercase letters present significant differences (*p* < 0.05).

**Figure 7 cells-11-03270-f007:**
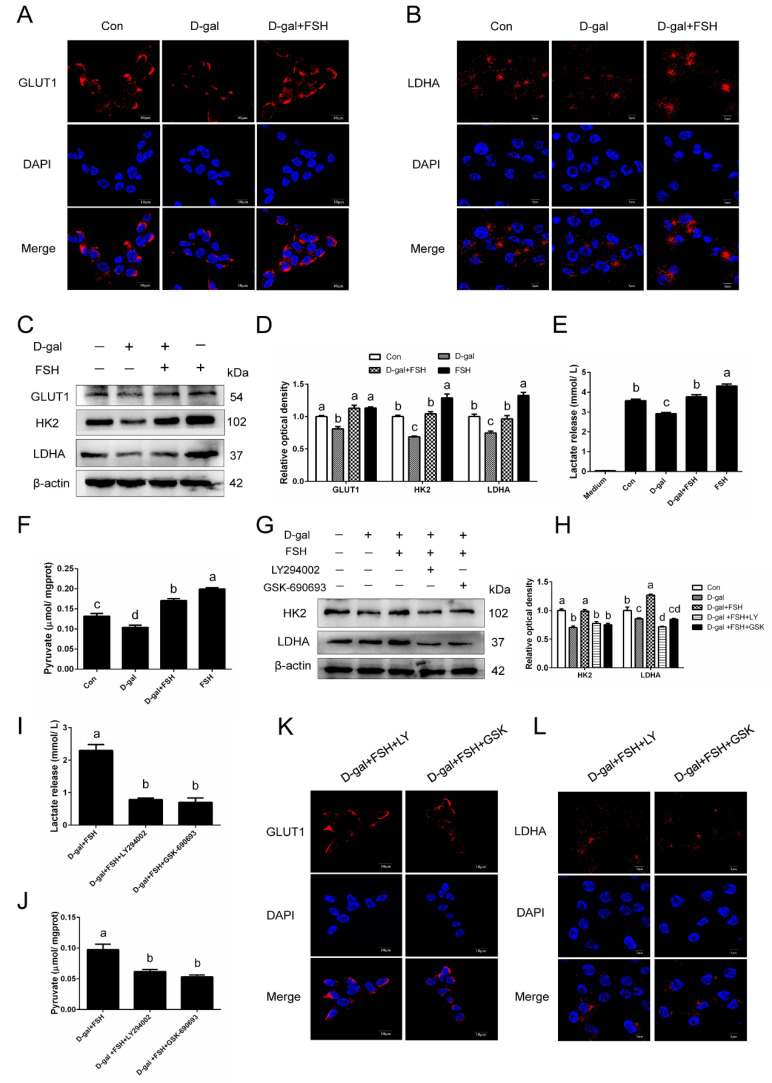
The enhanced effect of FSH on glycolysis in D-gal-induced senescent GCs. (**A**,**B**) The immunostaining of GLUT1 and LDHA in control cells, senescent cells (D-gal), and senescent cells under FSH treatment (D-gal + FSH). Scale bars: 10 μm and 5 μm. (**C**,**D**) Western blot of the glycolysis proteins of GLUT1, HK2, and LDHA in GCs exposed to D-gal for 24 h or treated with D-gal for 24 h and then cultured with FSH for another 24 h. (**E**,**F**) The pyruvate production and lactate release in GCs from the control, D-gal, D-gal + FSH, and FSH alone groups. (**G**,**H**) Levels of HK2 and LDHA in GCs treated with D-gal, D-gal + FSH, D-gal + FSH + LY294002, and D-gal + FSH + GSK690693 were detected by Western blotting. (**I**,**J**) Pyruvate production and lactate release shown in senescent GCs under FSH treatment, FSH + LY294002, and FSH + GSK690693. (**K**,**L**) Observation of GLUT1 and LDHA staining in senescent GCs that were treated as described above. Scale bars: 10 μm and 5 μm. Different lowercase letters present significant differences (*p* < 0.05).

**Figure 8 cells-11-03270-f008:**
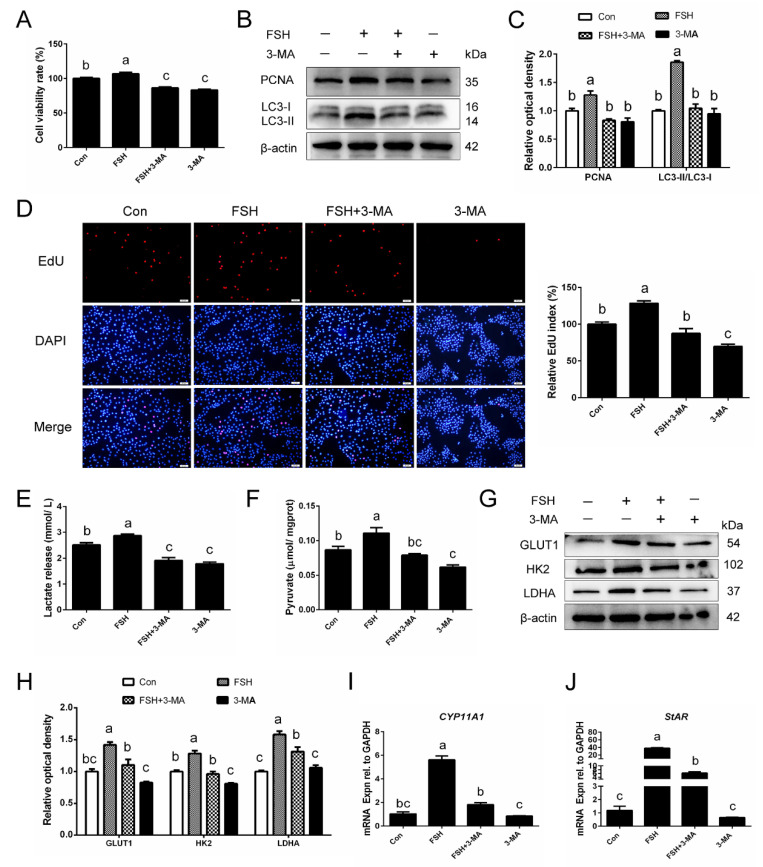
FSH-induced autophagy promotes naturally aged GC proliferation and glucose metabolism. (**A**) The GCs of D580 hens were incubated with or without FSH for 24 h and then treated with 3-MA for another 6 h. Cell viability detection using the CCK-8 assay (n = 8). (**B**,**C**) GCs treated with the indicated treatments were collected for examining the expression levels of PCNA and LC3 by Western blot. (**D**) The treated GCs were labeled with EdU, EdU-positive cells, red. Scale bar: 50 μm. (**E**,**F**) The detection of pyruvate and lactate in GCs that were treated as above. (**G**,**H**) The protein expression of GLUT1, HK2, and LDHA was detected by Western blotting. Relative protein levels were normalized to β-actin. (**I**,**J**) The function-related genes, *CYP11A1* and *StAR*, were determined by RT-qPCR. *GAPDH* was used as an internal control. Different lowercase letters present significant differences (*p* < 0.05).

**Figure 9 cells-11-03270-f009:**
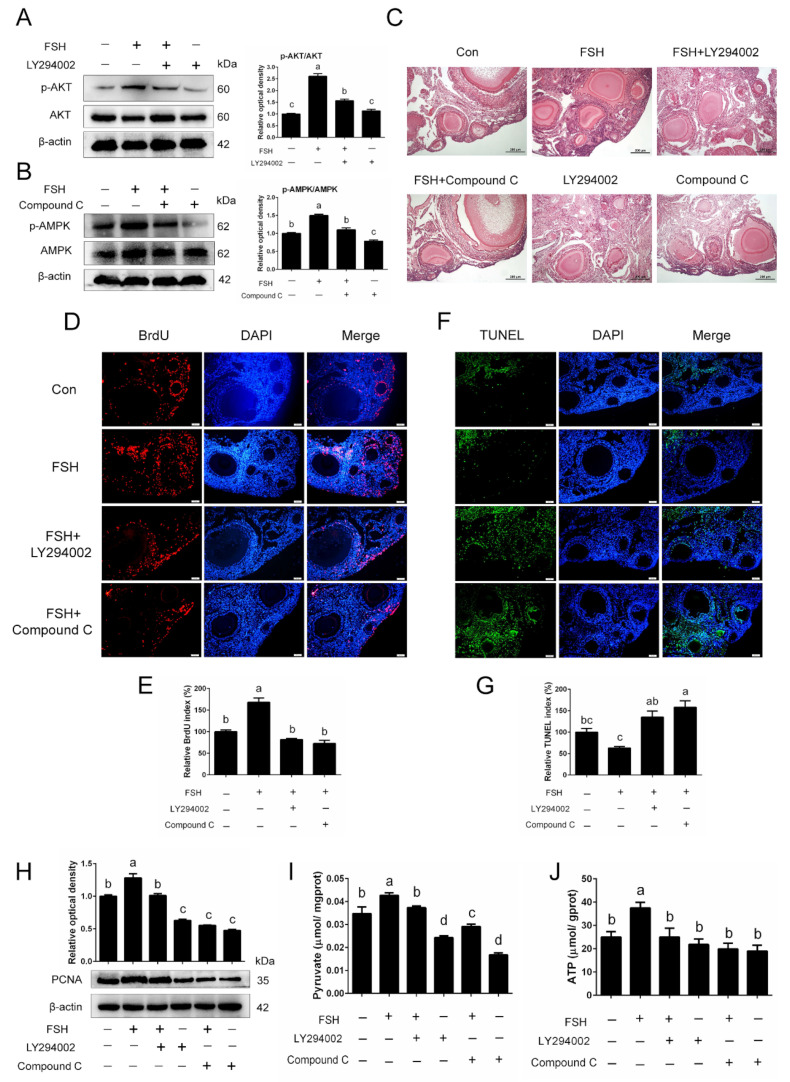
Improved effect of FSH on chicken ovarian aging was inhibited by the blockage of AKT and AMPK signaling. (**A**,**B**) Ovarian tissues of D580 hens were treated with FSH, FSH + LY294002, FSH + Compound C, LY294002, or Compound C for 72 h, respectively. The expression of p-AKT, AKT, p-AMPK, and AMPK was determined by Western blotting. (**C**) Representative H&E staining of ovaries. Scale bar: 200 μm. (**D**–**G**) BrdU incubation and TUNEL labeling of ovaries that were treated as indicated. Scale bar: 50 μm. (**H**) Level of PCNA in ovaries treated as indicated. (**I**,**J**) Pyruvate and ATP content of ovaries.

**Figure 10 cells-11-03270-f010:**
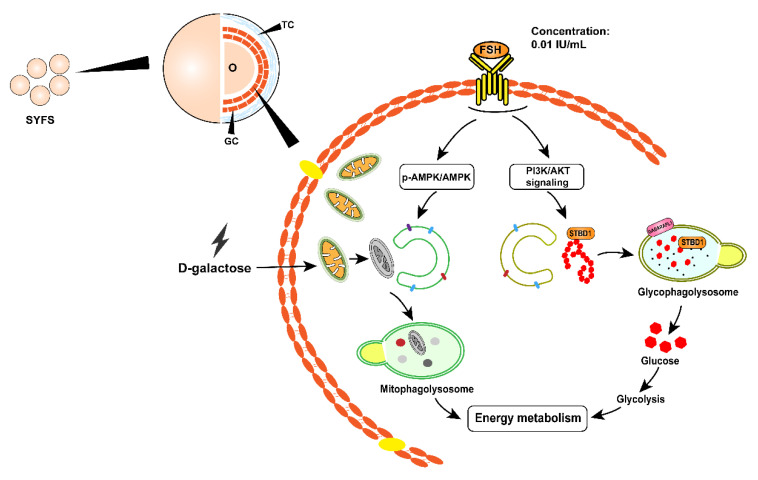
A proposed model for the protective effect of FSH in D-gal-induced premature senescent GCs of chicken. FSH regulates the mitophagic and glycophagic levels in isolated D-gal-induced premature senescent GCs by activating the AMPK and PI3K/AKT signaling pathways, thereby promoting increased energy metabolism.

**Table 1 cells-11-03270-t001:** Primers for PCR analysis.

Gene Name	Accession Number	Primer Sequence (5′-3′)
*Glut1*	NM_205209.1	GCAAGATGACAGCTCGCCTGTCTTCAATCACCTTCTGCGG
*HK1*	NM_204101.1	GGAGGATCAGGTCAAAAAGATCGTTCACTGTCGCTGTTGGGTT
*HK2*	NM_204212.1	GAAGGGTTTCAAAGCCACGGAGGTCAAACTCCTCTCTGCG
*PKM*	XM_015278796.2	CATGCAGCACGCTATTGCTCGTGGTGTACACTGTGGCGTA
*LDHA*	NM_205284.1	GAAGACGCCGGCAGTACAACCAACCACGCTGATCTTGT
*LDHB*	NM_204177.2	ACGTTATGGCGACCCTGAAGATCACAAAGACCCTTGCCGA
*CYP11A1*	NM_001001756.2	CCGCTTTGCCTTGGAGTCTGTGCGATGAACTGCTGTGCCTCTGG
*StAR*	NM_204686.2	AGGGTTGGGAAGGACACTCTGATCGGGAGCACCGAACACTCACAAAG
*GAPDH*	NM_204305.1	AGTCAACGGATTTGGCCGTACCGTTCTCAGCCTTGACAGT

**Table 2 cells-11-03270-t002:** Comparison of PK, LDH, and SDH levels in serum and ovaries from D280 and D580 hens.

Parameters	Sample	D280	D580
PK	Serum (U/L)	55.40 ± 4.746 ^a^	27.86 ± 6.704 ^b^
LDH	Ovary (U/g prot)	95.55 ± 8.076 ^a^	42.23 ± 8.325 ^b^
	Serum (U/L)	2070 ± 203.8 ^b^	2704 ± 163.4 ^a^
	Ovary (U/g prot)	4626 ± 479.5	4662 ± 410.2
SDH	Serum (U/mL)	142.2 ± 11.31 ^a^	104.4 ± 5.42 ^b^
	Ovary (U/mg prot)	21.42 ± 1.74 ^a^	11.85 ± 0.771 ^b^

Data are expressed as means ± SEM. Different lowercase letters indicate significant differences (*p* < 0.05).

## Data Availability

The raw data supporting the conclusions of this article will be made available by the authors, without undue reservation.

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
