# Peer review of "Follicle-Stimulating Hormone Alleviates Ovarian Aging by Modulating Mitophagy- and Glycophagy-Based Energy Metabolism in Hens"

_cells, 2022, doi:10.3390/cells11203270_

Round 1

Reviewer 1 Report

Comments:

Title: The statement from the title of the article «Follicle-Stimulating Hormone Alleviates Ovarian Aging by Modulating Mitophagy- and Glycophagy-Based Energy Metabolism in the Hens» is not proven by the data presented in it. The described experiments do not prove that FSH affects specifically aging. Ovarian cells from 580 day old hens do not turn into ovaries from 280 day old hens.

Line 13: There is no indication of what the abbreviation AMPK stands for.

Lines 306, 574: If all sections are numbered «1. Results» should be «3. Results» and «1. Discussion» should be «4. Discussion».

Lines 322, 352, 387, 446, 476, 508: In the results section, all items have the number "1.1".

Lines 332, 337: Grammar error - should be «found», not «founded».

Lines 142-152: It is worth clarifying how the composition of the culture medium changes when D-galactose is added. Is the medium highly diluted when the D-galactose solution is added or is it added as a powder?

Line 174: I assume that CCK8 assay and IDU assay are two different methods. It should be clearly stated in which cases which test was used, and these methods should be described separately. In the figures in the "Results" section (e.g. Fig 8A), the "cell viability" value is more than 100%, I assume that another indicator is measured in this test.

Figure 1 C – Six out of ten ovaries are displayed, please add rest of the data obtained to the Figure or change the n to six.

Figures starting from 3 and following (starting from line 352): the age of the hens from which the oocytes were taken is unclear

Line 367: It is doubtful that it is correct to use the beta-galactosidase assay to determine senescence by measuring beta-galactosidase activity when excess galactose is added to the culture medium.

Figure 2 F, 2 H – Please clarify the exact source of proteins for 3 presumably triplicate samples “D280…” and “D580…” – three extracts from one hen? Three hens? Three lanes with the same sample?

Figure 2 G, 2 I – please clarify the method of bands quantification and SD calculation. It seems that visible band intensities in Figure 2 H, antigen PFK2, samples D280 SYF, varies at least twofold and the SD (error bar) on the Figure 2 I, PFK2, white box, is around 7%.

Line 353. I think that both D280 and D580 GC’s should be treated by the D-Gal and tested for signs of ageing and senescence.

Fig 3 A and C: Galactose affects the osmolality of the medium - a hyperosmolal medium can itself be toxic. How physiological is the model that uses galactose at a final concentration of 200 mM? Typical concentration for glucose in the culture medium is about 10 mM, blood glucose 6 mM.

Fig 5  A, B: It is worth adding a control - cells without galactose treatment, but with FSH treatment. It is unclear whether the cells will stop dying. There is a possibility that they die not from galactose-induced aging, but from the absence of the stimulating effect of FSH.

Figure 8: It is not clear whether FSH affects the oocytes of old and young hens in the same way. It seems to me that treatments on oocytes taken from 580-day-old hens should be repeated for oocytes from 280-day-old hens. Fig. 8.A. The "cell viability" value is more than 100%, I assume that another indicator is measured in this test.

General Considerations:

1. There is also a lack of data on what endogenous FSH levels are in 280 and 580 day old hens.

2. Cellular senescence is a stable cell cycle arrest, complete cessation of cell division, the observed slowdown in cell division does not indicate that they go into senescence.

3. There is a lack of data on what signs of aging were observed in the model of galactose-induced cell aging, except for an increase in the activity of lysosomal galactosidase.

4. The fact that cells divide when FSH is added does not imply that FSH affected mitochondrial regulation.

Reviewer 2 Report

The work by Dong et al. demonstrated a role for follicle stimulating hormone (FSH) in regulating ovarian aging in hens. The authors used two different time points for young and aged hens and checked ovarian morphology, egg laying performance and senescence-related changes in ovarian energy metabolism by measuring the level and activity different glycolytic enzymes. The authors found that decreased ovarian glucose metabolism capacity is associated with the ageing process and such decrease was related to poor glycolysis in granulosa cells (GCs). Further, in cultured GCs, using D-galactose (D-gal), a reducing monosaccharide, that triggers excessive production of reactive oxygen species (ROS) and advanced glycation product (AGE) accumulation, the authors further documented that while D-gal lowers GCs viability but does not promote apoptosis of GCs, treatment of GCs with FSH significantly restored the loss of GC viability. More detailed analysis by the authors showed that FSH reversed D-gal-induced mitochondrial damage of GCs by promoting mitophagy and glycophagy via the AMPK pathway and PI3K/AKT pathways respectively. Moreover, the authors found that FSH increases glycolysis in of D-gal-treated GCs via PI3K/AKT pathway. Finally, the authors documented that enhanced effect of FSH on glycolysis in naturally aged GCs is mediated by PI3K/AKT and AMPK pathway, giving the work a more reliable in vivo context. 

While the work is extensive and critical for understanding the role of FSH-mediated signaling pathways that regulate ovarian aging, there are couple of experiments without any context, at least in the writing it is not clear. Additionally, this reviewer has couple of major comments regarding the level of FSH and FSHR in the young and aged hens.

Major comments:

1. What is the physiological serum concentration of FSH in early (D280) and aged (D580) hens? This is essential to make the argument that FSH indeed is lowered during the later laying stages.

2. What is the concentration of FSH receptor in the ovary in early (D280) and aged (D580) hens? This is essential to understand whether lowered FSH-mediated functions are due to the levels of receptor or the hormone itself.

3. L508: What is the rationale for checking autophagy rather than mitophagy in the in naturally aged GCs?

Minor comments:

L8: “hormone in the productive axis” should be “reproductive axis”

L11: mention granulosa cells for the first time use of GC

L51: Please reconstruct this sentence

L56: “LC3-I is lapidated to LC-II to” what is lapidated?

L148: what is the rationale for selecting the FSH doses? Are these doses physiological or supraphysiological according to the hen’s serum FSH level?

L317: Fig.1: marking F1-F6 in the histology would be better for the readers unfamiliar with this model.

L352: The authors checked GC viability using different cell proliferation markers and thus the point should be on that rather than “D-gal induces premature senescence”

L353: “the functions of GCs, we treated GCs” should mention ‘isolated from SYFs from ovaries of D280 or D580’?

L446: The logic of checking glycophagy in D-gal-induced premature senescent is not written here. It would be better if the authors give a little context for checking glycophagy.

L411: ‘robust mitochondrial injury’ sounds better than ‘serious mitochondrial injury’

L490: “To prove the conjecture” should be “To test this possibility”, otherwise it sounds like the authors knew the data and wanted to prove it.

L515: what is the function of 3-MA?

Round 2

Reviewer 1 Report

I received a detailed answer in which most of my comments were reasonably addressed. The manuscript was significantly improved, I would like to thank authors for this revision.

At the same time, one significant topic remains unchanged – the observed state of cells in the manuscript may be due to their senescence, or may only be similar to cellular senescence.

The formal definition of cell senescence is quite generally accepted – and includes «the absolute arrest of the cell cycle».

Below are some quotes:

«Cellular senescence is a state of permanent cell cycle arrest that was initially defined for cells grown in cell culture.» https://doi.org/10.1016/B978-0-12-394447-4.30066-9 https://www.sciencedirect.com/science/article/pii/B9780123944474300669

«Cellular senescence is an irreversible cell cycle arrest…»  https://doi.org/10.1016/B978-0-12-820071-1.00007-4

«Cellular senescence (CS) is a phenomenon characterized by the permanent cell growth arrest in normal and altered physiological processes.» https://doi.org/10.1016/B978-0-12-804274-8.00023-0

«Cellular senescence is a hallmark of aging that is characterized by irreversible cell cycle arrest in response to various stress stimuli.» https://doi.org/10.1016/B978-0-12-820071-1.00013-X

Therefore, the beta-galactosidase test seems to me insufficient to describe the state of the cell as the cellular senescence. I see that there are plenty of publications where this method is used to detect the activity of senescence-associated beta-galactosidase (SA-beta-gal), e.g. [Itahana K, Campisi J, Dimri GP. Methods to detect biomarkers of cellular senescence: the senescence-associated beta-galactosidase assay. Methods Mol Biol. 2007;371:21-31. doi: 10.1007/978-1-59745-361-5_3. PMID: 17634571.

Still I consider it incorrect to use a combination of the galactose-induced aging model and the beta-galactosidase assay as a method for detecting the state of cell senescence, despite the fact that many researchers use this model and this method separately. Exposure of the cell to a high concentration of galactose will lead to the activation of galactosidases, which may be erroneously considered "senescence-associated", not being such at all.

I consider it correct to use the term “cellular senescence” if there is more evidence obtained by different methods, for example, increased expression of a number of specific senescence markers, such as p16INK4A, p14ARF/p19ARF, and p21 [p16INK4A, p14ARF/p19ARF, and p21 [Wagner, K.-D.; Wagner, N. The Senescence Markers p16INK4A, p14ARF/p19ARF, and p21 in Organ Development and Homeostasis. Cells 2022, 11, 1966. https://doi.org/ 10.3390/cells11121966 ]; as well as activation of a senescence-associated secretory phenotype (SASP) - expression of the corresponding cytokines, chemokines, extracellular matrix-degrading proteins, and other factors. The senescence-related transcripts (Cdkn1a, Cdkn2a, Pai-1 and Hmgb1) are described for ovaries [Ansere VA, Ali-Mondal S, Sathiaseelan R, Garcia DN, Isola JVV, Henseb JD, Saccon TD, Ocañas SR, Tooley KB, Stout MB, Schneider A, Freeman WM. Cellular hallmarks of aging emerge in the ovary prior to primordial follicle depletion. Mech Ageing Dev. 2021 Mar;194:111425. doi: 10.1016/j.mad.2020.111425. Epub 2020 Dec 28. PMID: 33383072; PMCID: PMC8279026] which also could have homologues in hens.

I believe that this article contains a description of a large experimental work and can be published in one of two cases 1) if an explanation is added to it that the term "senescent cells" is interpreted by the authors more widely than is generally accepted, or 2) if the authors demonstrate real evidence that that the studied cells go into a state of senescence. Evidence must be different from the beta-galactosidase test and must be different from increased expression of genes that are upregulated in cells not only under senescence but also under metabolic stress (such as p53, p21). Senescent cells must lose their ability to divide in response to the mitogen. The studied cells restored the ability to divide after the addition of the mitogen - FSH. I believe the reason is that they were not senescent, but only in a state of metabolic stress.

Reviewer 2 Report

The authors reviewed the manuscript appropriately and this reviewer is satisfied. 

Author Response

Thank for your positive suggestions.